# Universal entanglement dynamics following a local quench

**Romain Vasseur[1,2] and Hubert Saleur[3,4⋆]**

**1** Department of Physics, University of California, Berkeley, California 94720, USA
**2** Materials Sciences Division, Lawrence Berkeley National Laboratory, Berkeley, California 94720, USA
**3** Institut de Physique Théorique, CEA Saclay, 91191 Gif Sur Yvette, France
**4** Department of Physics and Astronomy, University of Southern California, Los Angeles, California 90089-0484, USA

⋆ saleur@usc.edu

## Abstract

We study the time dependence of the entanglement between two quantum wires after suddenly connecting them via tunneling through an impurity. The result at large times is given by the well known formula $S(t) \approx \frac{1}{3} \ln t$. We show that the intermediate time regime can be described by a universal cross-over formula $S = F(t T_K)$, where $T_K$ is the crossover (Kondo) temperature: the function $F$ describes the dynamical "healing" of the system at large times. We discuss how to obtain analytic information about $F$ in the case of an integrable quantum impurity problem using the massless Form-Factors formalism for twist and boundary condition changing operators. Our results are confirmed by density matrix renormalization group calculations and exact free fermion numerics.



# 1   Introduction

It is well known [1] that the entanglement of two semi-infinite gapless spin chains initially separated and suddenly connected at time $t = 0$ grows logarithmically with time as $S = \frac{c}{3} \ln \frac{t}{a}$ where $a$ is a UV cut-off, and $c$ is the central charge of the conformal field theory (CFT) describing the low energy excitations of the chains, e.g. $c = 1$ for XXZ spin chains. This result is a cornerstone of the physics of local quenches, and has been studied and generalized in many contexts [2,3].

The logarithmic growth is only a large time behavior. Interesting dynamics can occur at intermediate times, and reveal much, in particular about the physics of quantum impurity problems. Indeed, it is possible to perform the quench in many different ways. An interesting variant involves two semi-infinite chains initially separated but suddenly connected at time $t = 0$ via *weak* tunneling through an extra site (a "dot"). This is equivalent, in the case of free fermions chains (which can be thought of as Fermi liquid leads) to a quench in the resonant level model (RLM) [4]. Adding an extra interaction between the dot and the wires [5] leads to a quench in the more general interacting RLM (IRLM). This model, in equilibrium, exhibits crossover physics similar to the physics of the Kondo model, with a weakly coupled two level system (the spin 1/2 impurity) at high-energy, a strongly coupled screened impurity at low-energy, and a crossover (Kondo) temperature $T_K$ [4].

Whenever the equilibrium physics exhibits such a crossover, time evolution is expected to exhibit the same features. In the IRLM model for instance, long times being equivalent to low-energy or long distances, the entanglement between the two halves of the system at large times should be determined by low-energy physics, where the impurity is *screened,* and the chain appears *healed* [6,7], exactly like in the problem we first discussed of a brutal quench to a homogeneous chain. Hence one expects $S \approx \frac{c}{3} \ln \frac{t}{a}$ for $t \gg T_K^{-1}$. At small times however, the chains should appear only weakly coupled, and the entanglement obviously must be much smaller. In fact, the entanglement in problems of this type is expected to admit a universal form in the limit where both the time and $T_K^{-1}$ become much larger than the bandwidth. We will argue soon that in this limit, one has

$$S = F_g(t T_K), \tag{1}$$

where $F_g(x)$ is a *universal function* (depending on the interaction parameter $g$ to be defined later; $g = \frac{1}{2}$ for the RLM), which should approach 0 (resp. $\frac{c}{3} \ln x$) in the limit of small (resp. large) value of the argument $x$ (this function $F$ was studied numerically in Ref. [8], see also *e.g.* [9–11] for related numerical studies).

While the large time logarithmic behavior can be obtained relatively easily using methods of conformal field theory [1], the crossover function $F$ is a complicated object, whose calculation requires a considerable effort, since it embodies the whole multi-scale physics of the problem, and involves a quantity – the entanglement – which is essentially non-local in terms of the original variables. We shall present results for general interactions obtained via numerical matrix-product state methods. In the RLM case, which is naively "non-interacting" but

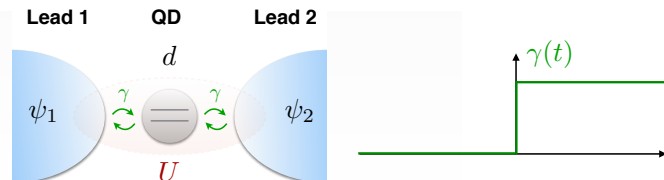

Figure 1: Quantum quench in the Interacting Resonant Level Model.

remains highly non trivial, we are able to perform an analytical calculation of $F$ thanks to the combined use of several form-factors (matrix elements [12]) approaches, relying on the integrability of the underlying quantum field theory. Our result – like those of similar calculations done in the past in equilibrium setups – rely on some steps that are not fully controlled.

## 2  The Interacting Resonant Level Model

### 2.1  Impurity model

The spinless IRLM involves two independent one-dimensional wires connected by tunneling through a quantum dot (the "impurity"). After unfolding the wires to represent them by chiral (say, right moving) fermions, the Hamiltonian reads

$$H = -i v_F \sum_{a=1,2} \int dx\, \psi_a^\dagger \partial_x \psi_a + \frac{\gamma}{\sqrt{2}} \sum_a \psi_a^\dagger(0) d + \text{h.c.} + U \sum_a \psi_a^\dagger \psi_a(0) \left( d^\dagger d - \frac{1}{2} \right). \quad (2)$$

Here, the label $a$ denotes the two wires, $\gamma$ is a tunneling amplitude (which we took, without loss of generality, to be the same for both wires), and $U$ is an interaction parameter with $d$ a fermion operator representing the degree of freedom on the dot. The equilibrium physics of the RLM ($U = 0$) is very simple, and applies to a broad class of systems including the anisotropic Kondo model at the Toulouse point or the problem of an impurity in a Luttinger liquid with parameter $g = \frac{1}{2}$. It is convenient to define $\psi_\pm = \frac{1}{\sqrt{2}}(\psi_1 \pm \psi_2)$, so that $\psi_-$ decouples from the impurity. The scattering matrix of the remaining fermion $\psi_+$ on the impurity then reads $S_+(\omega) = \frac{i\omega - T_K}{i\omega + T_K}$. The tunneling term is a relevant interaction, thus creating an energy scale $T_K = \frac{\gamma^2}{2}$, and the system flows under renormalization from the $\gamma = 0$ fixed point (independent wires) to a strong coupling fixed point $\gamma = \infty$ where the impurity is completely hybridized with the wires. At low energy, the only remaining effect of the impurity is a phase shift $\psi_+(0+) = -\psi_+(0-)$. When $U \neq 0$, the fermion scattering is more complicated, but the essential crossover phenomenon remains – note that $U$ corresponds to a marginal perturbation that modifies the critical properties continuously. The energy scale now varies as $T_K \propto \gamma^{1/(1-g)}$ where $g$ depends on $U$, with [7]

$$g = \frac{1}{4} + \frac{(U - \pi)^2}{4\pi^2}. \quad (3)$$

Note that $g \geq \frac{1}{4}$. The minimum is attained at the self-dual point [13, 14].

### 2.2  Quantum quench

We are interested in the quantum dynamics of this system after suddenly turning on the tunneling $\gamma$. Let $H_0 = H(\gamma = 0)$ be the Hamiltonian of the system for $t < 0$, and $H_1 = H(\gamma)$ the

Hamiltonian for $t \geq 0$ (see Fig. 1). The framework presented here is quite general and can be applied at finite temperature, but for simplicity, we will only consider the case $T = 0$ and imagine that the system is initially prepared in the groundstate $|\Psi(0)\rangle = \left|\psi_0^{(0)}\right\rangle$ of $H_0$ for $t < 0$. The wave function of the system at time $t$ is then $|\Psi(t)\rangle = e^{-iH_1 t} |\Psi(0)\rangle$, and the density operator is $\rho(t) = |\Psi(t)\rangle \langle \Psi(t)|$. We define a reduced density matrix by tracing over the right wire and the impurity (which we denote by system $B$), $\rho_A(t) = \mathrm{Tr}_B \rho(t)$. The entanglement entropy is then $S(t) = -\mathrm{Tr}[\rho_A(t) \ln \rho_A(t)]$. Our goal is to compute $S(t)$ as a function of the time, and the parameter $\gamma$. Since we are interested in the whole crossover of this function, perturbative approaches are bound not to be very successful [15, 16], and we turn to non-perturbative techniques.

## 3  Form factors

The first natural idea is to use the integrability of the model [5]. However, if the Bethe-ansatz allows control of many quantities in equilibrium, the study of non-equilibrium properties is more involved. This is especially true of the entanglement, which requires the use of several kinds of form-factors (FF). We shall illustrate the main ideas by discussing the case of the RLM. The FF approach in this case relies on the natural description of the Hilbert space as a Fock space of quasiparticle fermionic excitations, and uses the matrix elements of local operators, which are known thanks to vast, earlier and mostly axiomatic, considerations.

To be more precise, we first attempt to calculate, instead of $S(t)$, the Rényi entropy $\mathrm{Tr} \rho_A^N(t)$. As discussed in [1, 17, 18], such a trace can be calculated by introducing $N$ replicas of the system, with a "twist-operator" $\tau_N$ inserted to the immediate left of the origin: the role of this operator is to perform the partial trace over system $B$, while iterating $N$ times $\rho_A$. We then have

$$S_A(t) = -\frac{d}{dN} \langle \Psi(t)| \tau_N(x = -\epsilon) |\Psi(t)\rangle \bigg|_{N=1}, \qquad (4)$$

where we have assumed that $\tau_N$ is normalized by the one point function $\langle \Psi(0)| \tau_N(x = -\epsilon) |\Psi(0)\rangle$ at time $t = 0$, and $\epsilon$ is a regulator that we will send to zero at the end of the calculation. Note that now $|\Psi(t)\rangle \equiv \prod_{\alpha=1}^N |\Psi_\alpha(t)\rangle$ where $\alpha$ denotes the replicas. In all that follows, it is implied that all quantities (energies and bra/kets) refer in fact to the $N$ replicated theory. We do not mention this explicitly for ease of notation.

The first difficulty is of course that $|\Psi(0)\rangle$ is not an eigenstate of $H_1$. In order to determine $|\Psi(t)\rangle$ we need to introduce the basis of eigenstates of the Hamiltonian $H_1$, which we will denote by $\psi_1^{(n)}$ (with energy $E_1^{(n)}$) for the time being: the subscript 1 refers to $H_1$, and the upperscript $n$ labels the eigenstates. Hence we have

$$\langle \Psi(t)| \tau_N |\Psi(t)\rangle = \sum_{n,m} \left\langle \psi_0^{(0)} \middle| \psi_1^{(n)} \right\rangle e^{i(E_1^{(n)} - E_1^{(m)})t} \left\langle \psi_1^{(n)} \middle| \tau_N \middle| \psi_1^{(m)} \right\rangle \left\langle \psi_1^{(m)} \middle| \psi_0^{(0)} \right\rangle. \qquad (5)$$

We see that the determination of this quantity requires the knowledge of *two types of terms.* The overlaps between the ground state of $H_0$ and the excited states of $H_1$ are the "boundary conditions changing form-factors". They were initially determined in [19], and were used for instance in [20] to study the Loschmidt echo in the present quench. The matrix elements of the twist operator $\tau_N$ are the form-factors of the twist operators. They were determined in [21] and recently used for instance in [22] to study the crossover of the equilibrium entanglement entropy of a region surrounding the impurity with the rest of the system. Putting the two kinds of objects together presents new technical challenges, which we now briefly sketch in the RLM case ($U = 0$), although we emphasize that the same approach could in principle be applied to any integrable quantum impurity problem.

### 3.1 Crossover in the RLM

It is best to think of the eigenstates $\left|\psi_1^{(n)}\right\rangle$ in terms of elementary fermion excitations over the ground state. In presence of the impurity, there are two such excitations with energy $\omega \equiv e^\beta$, where $\beta$ is the rapidity:

$$
\begin{aligned}
&|\beta\rangle_{L,x>0} + r(\beta)|\beta\rangle_{R,x>0} + t(\beta)|\beta\rangle_{L,x<0}\,, \\
&|\beta\rangle_{R,x<0} + r(\beta)|\beta\rangle_{L,x<0} + t(\beta)|\beta\rangle_{R,x>0}\,,
\end{aligned}
\tag{6}
$$

where $r, t$ are simply related to the scattering matrix $S_+$ of the fermion $\psi_+$: $r(\omega) = \omega/(\omega + iT_K)$, $t(\omega) = iT_K/(\omega + iT_K)$. We check that when $T_K \to \infty$, $r \to 0$ and $t \to 1$, which corresponds to a healed chain, where left and right movers propagate without reflection.

While an infinity of processes contribute in principle, in practice it turns out that the form-factors expansion converges very fast, and only a few terms are necessary. The lowest order involves the following processes: $(a)$ a pair of particles is "created" at the first transition (that is, $\left|\psi_1^{(n)}\right\rangle$ involves two particles excitations over the vacuum) and destroyed by the twist (that is, $\left|\psi_1^{(n)}\right\rangle = \left|\psi_1^{(0)}\right\rangle$) $(b)$ a pair of particles is created by the twist and destroyed at the second transition $(c)$ a single particle is created at the first transition, is acted upon by the twist, and destroyed at the second transition. We used here the fact that $\tau_N$ can only create or destroy a pair of particles, while odd or even numbers of particles can be involved at the transitions (see appendix for a more complete discussion of this important aspect). Also, we recall that the presence of the $N$ replicas is implicit in the formulas and discussion. Hence, in processes $(a)$ and $(b)$, the pair can be created in the same or in different replicas, and in process $(c)$ the particle can be scattered into a different replica when acted upon by $\tau_N$.

These processes involve boundary conditions changing operators FF for both creation and destruction of a single particle, or of two particles. They involve twist FF for creation/destruction of pairs of particles, which are usually denoted by $F_2^{ij}(\beta_1, \beta_2)$ where $i, j = 1, \ldots, N$ label the different replicas. A crucial point is that, in all previous calculations [21,22], the important object was the *two point function* of twist operators, involving $|F_2|^2$. Here, in contrast, all terms at this order involve only $F_2$, that is, they crucially depend on the phase of the FF.

### 3.2 Leading form factor contribution

As often, the integrals over rapidities of particles involved in the processes are divergent at low-energy. This "IR catastrophe" is typical of the massless particle approach, and is easily taken care of by considering instead the derivative of $S$ with respect to time. One finds in the end the leading contribution:

$$
t\frac{\partial}{\partial t}S = \frac{tT_K}{2\pi} \int_0^\infty \frac{d\nu_1}{\nu_1^{1/4}} \frac{d\nu_2}{\nu_2^{1/4}} \frac{(\nu_1 - \nu_2)^2}{(\nu_1^2+1)(\nu_2^2+1)(\nu_1+\nu_2)^2} e^{\varphi(\ln \nu_1)+\varphi(\ln \nu_2)} \sin[tT_K(\nu_1+\nu_2)] + \ldots,
\tag{7}
$$

with

$$
\varphi(x) = \int_{-\infty}^{+\infty} \frac{dy}{4y}\left(\frac{2}{y} - \frac{\cos\frac{x}{2\pi}y}{\cosh\frac{y}{4}\sinh\frac{y}{2}}\right).
\tag{8}
$$

In the large time (IR) limit $tT_K \gg 1$, this gives $t\frac{\partial}{\partial t}S = \frac{1}{4}$, whereas the exact amplitude from CFT should be $\frac{c}{3}$ with $c = 1$ the central charge. In a way similar to equilibrium FF calculations [2, 21–23], we expect this amplitude to be corrected by higher-order FF contributions. We note that even at lowest order in the FF expansion, there are other contributions that were

not included in eq. (7). These contributions are subleading in the sense that they vanish in the IR limit $t T_K \gg 1$. They are generically hard to evaluate numerically (see supplementary material), but we checked that they remain relatively small throughout the whole crossover for the points where we were able to evaluate them.

While equation (7) is not exact, it is accurate numerically all over the crossover region, as we shall illustrate below, provided we perform a "brutal" renormalization of (7) by a factor $\frac{4}{3}$ to obtain the correct IR limit. Similar renormalizations have been used in equilibrium calculations in the past (see [16, 22]), and even though this procedure remains unpleasant, it can be checked in these simpler cases that going to higher orders does not modify significantly the first order FF contribution once properly renormalized (*i.e.* the higher order terms mostly give a "multiplicative" factor to the first order term). See supplementary material for more detail.

Of course, the FF formalism can in principle be extended to the more general IRLM problem. In this case however, the necessary expressions for the matrix elements of the twist operators and for the boundary interaction changing operators have not been entirely worked out. As we shall see, the essential qualitative aspects are already present in the RLM, so we turn simply to numerical calculations.

## 4 Lattice model

### 4.1 Numerics

We now turn to numerical results to study the full crossover in the IRLM and to validate the FF approach in the RLM case. We consider a lattice version of the IRLM

$$H = -J \sum_{a=1,2} \sum_{i=1}^{L-1} (c_{i+1}^{a\dagger} c_i^a + \text{h.c.}) - J' \sum_a (d^\dagger c_1^a + \text{h.c.}) + U_l \sum_a \left( d^\dagger d - \frac{1}{2} \right) \left( c_1^{a\dagger} c_1^a - \frac{1}{2} \right), \quad (9)$$

with $J = 1$ so that the Fermi velocity is $v_F = 2$, and where the $c$ and $d$ fermions correspond to the gapless leads and the dot degree of freedom, respectively. At sufficiently low energies $J' \ll J = 1$, the system is described by the effective field theory (2), with $\gamma \propto J'$ and $U \sim U_l$ (the precise relation between $U_l$ and $U$ is non-universal). In the non-interacting RLM case, it is even possible to identify exactly the energy scale $T_K = 2J'^2/J$ including non-universal $\mathcal{O}(1)$ factors, by computing for example the transmission probability both from (2) and (9) [14].

We determine the entanglement following a quench from $J' = 0$ to $J' \neq 0$ using the time evolving block decimation (TEBD) algorithm [24] and a fourth order Trotter decomposition with time step $dt = 0.1$, increasing the dimension of the matrix product state to keep the discarded weight below $10^{-7}$ throughout the unitary time evolution. The initial state with leads of size $L = 256$ (total system size $N = 513$) is determined using standard density-matrix renormalization group (DMRG) techniques [25, 26]. In the RLM case, the entanglement can also be obtained by diagonalizing the fermion correlation functions $\langle c_i^\dagger(t) c_j(t) \rangle$ [27, 28]. In this case, we compute the entanglement for leads with $L = 500$ sites ($N = 1001$) for different values of $J'$, and find that the results indeed collapse onto a universal curve after rescaling the time scale by a factor $T_K$. We note that whereas one wishes to have $J'$ as small as possible in order to describe accurately the field theory limit, the finite-size effects are stronger when $J'$ is small so the range of values for $J'$ must be chosen carefully.

For $U = 0$, the determination of the entanglement from (7) requires numerical evaluation of integrals with a strongly oscillating term at large-energy. In other calculations, this difficulty can be circumvented by going to imaginary time: this is not possible here, because the Heisenberg evolved operator $\tau_N(t)$ involves exponentials with a $\pm$ sign, and would make the integrals in imaginary time undefined. Calculation in real time is possible with a bit of care,

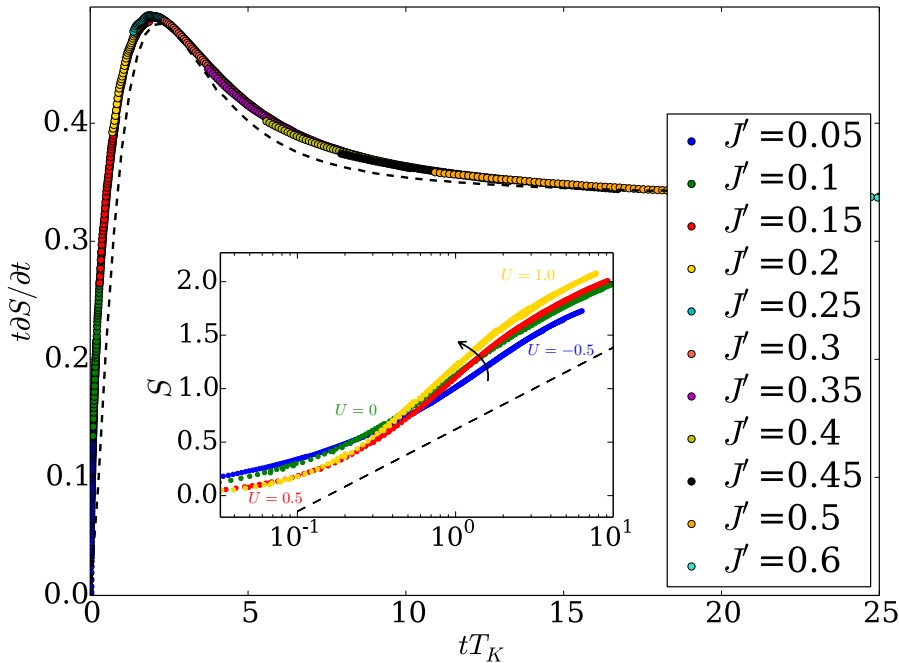

Figure 2: The instantaneous slope of the entanglement for $U = 0$, with the dashed line corresponding to the leading contribution FF calculation eq. (7). Inset: the entanglement itself, for various values of $U$ (for clarity, the numerical data for different values of $J'$ here carry the same color).

and we find the results shown on Fig. 2. The (only numerical) results for $U \neq 0$ are shown in the inset of Fig. 2, where we directly represented $S$ instead of the derivative (see also [8]).

Note that because of finite size effects, one expects the curves for small values of $J'$ to describe well the universal curve for small $t T_K$ only. We find that the FF expansion is in good agreement with our numerical results (though unfortunately not as good as in equilibrium calculations), even in the interesting non-perturbative region $t \sim T_K^{-1}$ where $S(t)$ has a non-trivial behavior – note that there is no free parameter in the results, which must match without possible rescaling of the time axis.

## 4.2 IR expansion

We see that curves for different values of $U$ are roughly similar: the maximum of $\partial S / \partial \ln t$ increases in the repulsive regime $U > 0$, and decreases in the attractive regime $U < 0$. It is tempting to investigate whether some of the shape of these curves can be recovered using perturbation theory. For the very far IR for instance, the system is essentially healed and the logarithmic result for $S(t)$ holds [1]. At large but finite times, the system appears almost healed, and can be described by a perturbed CFT. The leading perturbation in this case is proportional to the stress energy tensor, $H = H_{IR} - \frac{1}{\pi T_K} T + \ldots$. All other terms are known in principle, and one can attempt a perturbative calculation of the Rényi entropies - and thus the entanglement - following the lines of [14]. This gives a series in $1/(t T_K)$, whose leading term is $t \frac{\partial S}{\partial t} = \frac{1}{3} \left( 1 + \frac{4}{\pi^2} \frac{1}{t T_K} + \ldots \right)$. Higher order terms are difficult to calculate. Moreover – like in the equilibrium crossover discussed in [16], the resulting series only describes reliably the very deep IR regime, and cannot really be compared with numerical simulations. We notice, however, that the sign of the leading term indicates an approach to the CFT result from above,

as seen numerically or with the FF solution.

## 5 Discussion

This work is an important new step in our understanding of non-equilibrium quench dynamics of quantum impurity problems. In the RLM case, we have seen that the combination of two kinds of FF (and calculation in real time) can be successfully implemented, and gives results in good agreement with independent numerical studies of an equivalent lattice model. Like in other problems (see e.g. [22]), the FF calculation being carried out only at the lowest order, and a final renormalization of the result was required to get this agreement. It is not entirely understood at the present time why this renormalization works so well: it would be very interesting (but extremely tedious) to investigate higher orders to shed light on this question – see *e.g* [16] for a simpler, equilibrium example for which this can be done explicitly. Contrary to equilibrium cases, we also had to isolate a leading contribution that does not vanish in the IR among the lowest order terms: the other terms are "subleading" — they remain relatively small in the crossover regime — but they seem to be large enough to worsen the agreement in Fig. 2. The corresponding integrals are unfortunately hard to evaluate numerically and require additional regularizations (see supplementary material): more work would be needed to evaluate higher-order terms and clarify the importance of these FF contributions. In the more general IRLM case, we have found excellent scaling, confirming the idea that the time dependent properties are universal, and reflect the physics of the equilibrium RG flow.

While we have focussed on $t\frac{\partial S}{\partial t}$ in the FF approach for technical reasons, we stress that the entanglement itself is – as checked numerically – a universal function of $tT_K$. This can be established as follows. Observe that the one point function of the twist operator must take the form $\langle \tau \rangle \sim t^{-\alpha} f(tT_K)$. As in [22], this leads to $t\frac{\partial S}{\partial t}$ being a scaling function of $tT_K$. The point is now that we can integrate with respect to $t$ to get $S$ also as a scaling function, since we know the initial condition $S(t=0)=0$, and this is in contrast with the equilibrium case [22], where $S$ itself was affected by terms depending on $aT_K$ with $a$ the lattice spacing.

Plotting the derivative emphasizes however the intriguing fact that the instant slope (wrt $\ln t$) of the entanglement growth saturates at values *greater* than $\frac{c}{3}$ in the intermediate regime. Even though there is no general monotonicity requirement for this quantity, it is not totally clear what this means physically — but suggests that in a quasiparticle approach [1], the particles emitted after the quench carry an amount of entanglement that depends on their momentum, leading to a "crowding effect" before the CFT regime settles in. We also note that the entanglement in this quench behaves somewhat similarly to the logarithm of the Loschmidt echo, adding another example where these two quantities are qualitatively related [29].

In conclusion, we also note that it is possible to consider a quench between two different systems that both involve a non zero coupling $\gamma$ to the dot. In this case, the entanglement does not grow logarithmically, but saturates at large times. We have not been able to extract convincing scaling curves from the numerics in this case, and refrain from discussing it in more detail. We note however that it is easy to calculate the *difference* of entanglement between the two systems: one finds

$$S(T_K^{(1)}) - S(T_K^{(2)}) = \frac{1}{6}\ln(T_K^{(1)}/T_K^{(2)}). \tag{10}$$

Note now that this difference is much simpler than the logarithm of the overlap between the ground states of the two system [30, 31], which is a highly non-trivial function of $T_K^{(1)}/T_K^{(2)}$.

## Acknowledgments

We thank S. Haas and A. Roshani for useful discussions. This work was supported by the US Department of Energy (grant number DE-FG03-01ER45908) and the Advanced ERC Grant NUQFT (H.S.), and the Quantum Materials Program at LBNL (R.V.). We thank O. Castro Alvaredo and B. Doyon for working out some of the twist form-factors expressions needed in our calculation.

## A    Supplemental Material

### A.1    The calculation in the conformal case

In this supplemental material, we first provide more details regarding the FF calculations. For simplicity, we start with the conformal limit $T_K = 0$. We need the boundary conditions changing FF, which in this limit read simply, for R moving particles

$$G(\beta_2, \beta_1) = i \tanh \frac{\beta_{21}}{2}. \tag{11}$$

where $\beta_{21} = \beta_2 - \beta_1$, and

$$G(\beta_2, \beta_1) = \frac{{}_1\langle \beta_2, \beta_1 \,|0\rangle_0}{{}_1\langle 0\,|0\rangle_0}. \tag{12}$$

The FF for the creation of a single particle is obtained by letting the rapidity of the other particle go to $-\infty$ (so its moment and energy vanish), $G(\beta) = \pm i$ (the sign does not matter). Some of the necessary FF for the twist operator can be found in [21], in particular

$$\frac{1}{\langle \tau \rangle} \frac{d}{dn} \sum_{i=1}^{N} F_2^{\tau|ii}(\beta_1, \beta_2)\bigg|_{n=1} = \frac{i\pi}{2} \frac{\tanh(\beta_{12}/2)}{\cosh(\beta_{12}/2)}, \tag{13}$$

where $F \equiv \langle 0| \tau_N |\beta_1, \beta_2\rangle$. Our problem requires however the knowledge of other sums which were not considered before [32]:

$$\frac{1}{\langle \tau \rangle} \frac{d}{dN} \sum_{i,j=1}^{N} F_2^{\tau|ij}(\beta_1, \beta_2)\bigg|_{n=1} = \frac{i\pi}{2} \frac{\tanh(\beta_{12}/2)}{\cosh(\beta_{12}/2)} + \frac{\pi}{2\cosh^2(\beta_{12}/2)}. \tag{14}$$

The two particle contributions can then be organized as follows.

- (a) First process, $i \neq j$:

$$2 \times \int \frac{d\beta_1}{2\pi} \frac{d\beta_2}{2\pi} \frac{1}{2!} \sum_{i \neq j} F_2^{\tau|ij}(\beta_1, \beta_2) g^2 e^{-it(e^{\beta_1} + e^{\beta_2})}, \tag{15}$$

  Here, the factor 2 comes from the existence of L and R channels. $g$ is the (pure phase) one particle form-factor, $g = \pm i$. Replacing by the expression for the limit of the derivative $\frac{d}{dN}$ and factoring out $\langle \tau \rangle g^2$ we get

$$\int \frac{d\beta_1}{2\pi} \frac{d\beta_2}{2\pi} \frac{\pi}{2\cosh^2(\beta_{12}/2)} e^{-it(e^{\beta_1} + e^{\beta_2})}. \tag{16}$$

We go to new coordinates $x \equiv \frac{\beta_1 + \beta_2}{2}$ and $y \equiv \frac{\beta_1 - \beta_2}{2}$. This gives

$$\frac{1}{4\pi} \int dx\, dy \frac{e^{-2ite^x \cosh y}}{\cosh^2 y}. \tag{17}$$

It is convenient to calculate (here and below) the derivative wrt $t$ of this expression. The integral can then be done straightforwardly (an imaginary part must be added to $t$ to make it converge) and one finds, after re-integrating, the first contribution as $-\frac{1}{2\pi}g^2 \ln t$.

- (b) Second process, $i \neq j$:

  This gives immediately the conjugate: $-\frac{1}{2\pi}(\bar{g})^2 \ln t$.

- (c) Third process:

  In this case, we can lump together the cases $i = j$ and $i \neq j$. The fact that we have a particle destroyed and one created leads to a factor $g\bar{g}$, and we have, factoring it out together with $\langle \tau \rangle$:

$$2 \times \sum_{i,j=1}^{N} \int \frac{d\beta_1}{2\pi} \frac{d\beta_2}{2\pi} e^{-it(e^{\beta_1} - e^{\beta_2})} F_2^{\tau|ij}(\beta_1, \beta_2 - i\pi). \tag{18}$$

The form factor does not exhibit any pole when $\beta_{12} = 0$. Note that there is no symmetry factor $1/2!$ any longer, because $\beta_1$ is the rapidity of the created particle, $\beta_2$ the one of the destroyed particle, and these are distinguishable. The overall $2\times$ factor as usual comes from L and R channels. Going to the variables $x, y$ gives the contribution

$$\frac{1}{4\pi} \int dx\, dy \frac{1}{\cosh^2 y/2} e^{-2ite^x \sinh y}, \tag{19}$$

and a few easy manipulations lead to the contribution $-\frac{g\bar{g}}{\pi} \ln t$.

- (a')(b') Finally, we must come back to processes (a) and (b) when the particles created (or destroyed) are in the same replica. In this case indeed, we need an additional exchange contribution in the FF (and note the symmetry factor $1/2!$), and we find

$$2 \times \sum_{i=1}^{N} \int \frac{d\beta_1}{2\pi} \frac{d\beta_2}{2\pi} \frac{1}{2!} \times (i \tanh \frac{\beta_{12}}{2}) e^{-it(e^{\beta_1} + e^{\beta_2})} F_2^{\tau|ii}(\beta_1, \beta_2), \tag{20}$$

together with its complex conjugate. An easy calculation like before gives the contribution (after adding the complex conjugate) $-\frac{1}{4} \ln t$.

Adding up all these contributions, we find

$$S \approx \left(g^2 + (\bar{g})^2 + 2g\bar{g}\right) \frac{1}{2\pi} \ln t + \frac{1}{4} \ln t. \tag{21}$$

Recalling now that $g = i$, we see the first factor vanishes entirely, and we obtain simply

$$S \approx \frac{1}{4} \ln t. \tag{22}$$

We note that the correct amplitude should be $\frac{c}{3} = \frac{1}{3}$. The necessary correction would be provided – like in other FF calculations – by consideration of higher order terms. This is illustrated below in the case of a quench between two weakly connected chains. In most cases however - and the present problem is no exception - it is enough to impose the same renormalization $1/4 \rightarrow 1/3$ in the non conformal case as well. That is, to get the crossover expression (7) we consider the same processes and multiply the final result by 4/3 [22]. The origin of this procedure dates back to early works on massless form-factors (or the UV limit of form-factors for ordinary massive theories), in particular the work [33]. In many problems,

the integrals over rapidities involved in these form-factor calculations are divergent, *and* the sum over form-factor contributions, once the integrals are made finite by the introduction of a cut-off, is also divergent. These divergences can be controlled by calculating, instead of the quantity of interest (say a correlation function at distance $r$, which depends on the value of the crossover temperature $T_K$), the ratio of this quantity to the same quantity evaluated in the conformal case, and manipulating this ratio formally to cancel divergences. Put slightly differently, the form-factors program, in the massless case, seems better adapted at calculating ratios of quantities to their values in the conformal case. This is another way to interpret the renormalization we have carried out in this paper (and in [22]), even though, in the case at hand, we did, in fact, regulate divergences by taking a derivative w.r.t. time, and the form-factors series in fact does converge. Clearly, more work is needed to fully understand the role of higher-order contributions.

The $\frac{\partial}{\partial t}$ trick should not hide the fact that the integrals are initially IR logarithmically divergent. For instance, the last process leads to the integral

$$\int_{-\infty}^{\infty} d\beta_1 d\beta_2 \frac{(e^{\beta_1} - e^{\beta_2})^2}{(e^{\beta_1} + e^{\beta_2})^3} e^{\beta_1/2} e^{\beta_2/2} e^{-it(e^{\beta_1} + e^{\beta_2})}, \tag{23}$$

and a change of variables gives then, up to numerical factors

$$\int_0^{\infty} \frac{d\rho}{\rho} \int_{-\pi/4}^{\pi/4} \frac{d\phi}{\sqrt{\cos 2\phi}} \frac{(\sin \phi)^2}{(\cos \phi)^3} e^{-it\sqrt{2}\rho \cos \phi}, \tag{24}$$

and the $\rho$ integral is clearly logarithmically divergent. Meanwhile, applying $t\frac{\partial}{\partial t}$ gives a finite result where $t$ vanishes:

$$\int_{-\pi/4}^{\pi/4} \frac{d\phi}{\sqrt{\cos 2\phi}} \frac{(\sin \phi)^2}{(\cos \phi)^3} = \frac{\pi}{2}. \tag{25}$$

## A.2 General case

The non-conformal case is more complicated. While the processes and the twist form-factors expressions are the same, the boundary interaction changing form-factors have considerably more complicated expressions:

$$G(\beta_2, \beta_1) = -\frac{1}{4} \left( \frac{T_K^2}{e^{\beta_1} e^{\beta_2}} \right)^{1/4} \tanh \frac{\beta_{12}}{2} \frac{e^{\beta_1} + iT_K}{e^{\beta_1} - iT_K} \frac{e^{\beta_2} + iT_K}{e^{\beta_2} - iT_K} \Phi(\beta_1 - \beta_K) \Phi(\beta_2 - \beta_K), \tag{26}$$

where $T_K \equiv e^{\beta_K}$ and

$$\Phi(x) = \frac{1}{\cosh\left(\frac{x}{2} - \frac{i\pi}{4}\right)} \exp[\varphi(x)], \tag{27}$$

with $\varphi(x)$ given by (8). The conformal case is recovered when $T_K \to \infty$. In this limit

$$\Phi(\beta - \beta_K) \approx 2e^{-i\pi/4} e^{(\beta - \beta_K)/4}. \tag{28}$$

Also, since now the reflection coefficient $r$ is non zero, the channels split, and the change of boundary interaction can lead to the creation of pairs of R movers or pairs of L movers with different, $r$ and $t$ dependent amplitudes (the processes where one pair of R and one pair of L is created do not participate at this order, since only the RR and LL FF of the twist operator are non zero).

We start with the last processes $(a')(b')$ studied in the conformal case — that is, those involving a pair of particles created or destroyed at the transition *within the same replica*.

These should still be the dominating ones even if $T_K$ is finite. First, we replace the $i \tanh \beta_{21}/2$ by the more complicated expression $G(\beta_2, \beta_1)$. Second, we now must consider the asymptotic states and where the $\tau$ operator is inserted. If for instance it is inserted at $x > 0$ (and small so we do not have additional phase factors coming from the momentum), then if the transition created a pair of "L" asymptotic states, $\tau$ can destroy it with $\tau_L$, or can destroy its R moving reflected image with $\tau_R$: clearly, this changes the factor of two into a combination

$$1 + 1 \rightarrow 1 + r(\beta_1)r(\beta_2) - t(\beta_1)t(\beta_2). \tag{29}$$

So for instance we obtain, instead of (20)

$$\sum_{i=1}^{n} \int \frac{d\beta_1}{2\pi} \frac{d\beta_2}{2\pi} \frac{1}{2!} \, G(\beta_2, \beta_1) [1 + t(\beta_1)t(\beta_2) - r(\beta_1)r(\beta_2)] \, e^{-it(e^{\beta_1} + e^{\beta_2})} F_2^{\tau|ii}(\beta_1, \beta_2), \tag{30}$$

where the minus sign occurs because both $G$ and $F_2$ switch signs when one exchanges L for R. Note that this would be the only term if we started from already weakly connected chains.

We check that at large times we expect the integral to be dominated by rapidities going to $-\infty$ so $e^{\beta}$ is small. In this limit, $t \approx 1$ and $r \approx 0$ so we recover the result of the conformal quench. Meanwhile, at very small times we expect instead the region $r \approx 1$ to dominate, with a very small entanglement.

To proceed we need the expressions (where $u_i \equiv e^{\beta_i}$):

$$G(u_2, u_1) = -\frac{1}{4} \left( \frac{T_K^2}{u_1 u_2} \right)^{1/4} \frac{u_1 - u_2}{u_1 + u_2} \frac{u_1 + iT_K}{u_1 - iT_K} \frac{u_2 + iT_K}{u_2 - iT_K} \, \Phi_1(\ln \frac{u_1}{T_K}) \, \Phi_1(\ln \frac{u_2}{T_K}), \tag{31}$$

and

$$1 + t(\beta_1)t(\beta_2) - r(\beta_1)r(\beta_2) = iT_K \frac{u_1 + u_2 + 2iT_K}{(u_1 + iT_K)(u_2 + iT_K)}. \tag{32}$$

Meanwhile,

$$\frac{d}{dN} \sum F_2^{\tau|ii}(\beta_1, \beta_2) \Big|_{N=1} = i\pi \sqrt{u_1 u_2} \frac{u_1 - u_2}{(u_1 + u_2)^2}. \tag{33}$$

Hence, for process $(a')$ we have to consider the integral

$$-\frac{1}{32\pi} T_K \int_0^\infty \frac{du_1}{u_1} \frac{du_2}{u_2} \frac{u_1 + u_2 + 2iT_K}{(u_1 + iT_K)(u_2 + iT_K)} \left( \frac{T_K^2}{u_1 u_2} \right)^{1/4} \frac{u_1 - u_2}{u_1 + u_2} \frac{u_1 + iT_K}{u_1 - iT_K} \frac{u_2 + iT_K}{u_2 - iT_K}$$
$$\times \Phi(\ln \frac{u_1}{T_K}) \Phi(\ln \frac{u_2}{T_K}) e^{-it(u_1 + u_2)} \sqrt{u_1 u_2} \frac{u_1 - u_2}{(u_1 + u_2)^2}. \tag{34}$$

Rewriting $\Phi$ as in (27) we get

$$-\frac{i}{8\pi} T_K^{5/2} \int_0^\infty \frac{du_1}{u_1^{1/4}} \frac{du_2}{u_2^{1/4}} \frac{u_1 + u_2 + 2iT_K}{(u_1^2 + T_K^2)(u_2^2 + T_K^2)} \frac{(u_1 - u_2)^2}{(u_1 + u_2)^3}$$
$$\times \exp[\varphi(\ln \frac{u_1}{T_K})] \exp[\varphi_1(\ln \frac{u_2}{T_K})] e^{-it(u_1 + u_2)}. \tag{35}$$

Recall that, to get the physical contribution, we need to add the complex conjugate process $(b')$. We thus end up with two contributions

$$-\frac{1}{4\pi} T_K^{5/2} \int_0^\infty \frac{du_1}{u_1^{1/4}} \frac{du_2}{u_2^{1/4}} \frac{1}{(u_1^2 + T_K^2)(u_2^2 + T_K^2)} \frac{(u_1 - u_2)^2}{(u_1 + u_2)^2}$$
$$\times \exp[\varphi(\ln \frac{u_1}{T_K})] \exp[\varphi(\ln \frac{u_2}{T_K})] \sin[t(u_1 + u_2)], \tag{36}$$

and

$$\frac{1}{2\pi} T_K^{7/2} \int_0^\infty \frac{du_1}{u_1^{1/4}} \frac{du_2}{u_2^{1/4}} \frac{1}{(u_1^2 + T_K^2)(u_2^2 + T_K^2)} \frac{(u_1 - u_2)^2}{(u_1 + u_2)^3}$$

$$\times \exp[\varphi(\ln \frac{u_1}{T_K})] \exp[\varphi(\ln \frac{u_2}{T_K})] \cos[t(u_1 + u_2)]. \quad (37)$$

While the first contribution is convergent at low energy, the second contribution exhibits a logarithmic divergence – the same divergence we encountered in the conformal case. Note however that in the conformal case we brutally set $T_K = 0$ (and thus had no cutoff left) while now, if $T_K \to \infty$, it is natural to rescale all the variables, and end up with a function of $t T_K$.

We observe that if we apply $\frac{t\partial}{\partial t}$ to our expressions, we will now get something that is convergent, and for which we can shift the variables $\beta_i$ (rescale the variables $u_i$) so in the end we get a function of $t T_K$ only. After the usual sign switch to get the entanglement we find finally the contributions from processes $(a'), (b')$:

$$t\frac{\partial}{\partial t} S^{(a')+(b')} = \frac{t T_K}{2\pi} \int_0^\infty \frac{dv_1}{v_1^{1/4}} \frac{dv_2}{v_2^{1/4}} \frac{1}{(v_1^2 + 1)(v_2^2 + 1)} \frac{(v_1 - v_2)^2}{(v_1 + v_2)} \exp[\varphi(\ln v_1)] \exp[\varphi(\ln v_2)] \times$$

$$\left\{ \frac{\sin[t T_K(v_1 + v_2)]}{v_1 + v_2} + \frac{1}{2} \cos[t T_K(v_1 + v_2)] \right\}, \quad (38)$$

with $v_i = u_i/T_K$. We now go back to the other processes $((a), (b), (c))$ whose contribution summed up to zero in the conformal case. We will organize the contributions in the non conformal case similarly. The $(a)$ and $(b)$ processes correspond again to pairs of particles being created or destroyed, but this time on different replicas. This means that, on the one hand, we get a product of $G$ factors corresponding to the creation of a single particle on a given replica, and also we get the $F_2^{\tau|i \neq j}$ term for the action of the twist field $\tau$. After a straightforward calculation, we find the corresponding term to be:

$$t\frac{\partial}{\partial t} S^{(a)+(b)} = -\frac{t T_K}{\pi} \int_0^\infty dv_1 dv_2 \frac{(v_1 v_2)^{1/4}}{(v_1^2 + 1)(v_2^2 + 1)} \exp[\varphi(\ln v_1)] \exp[\varphi(\ln v_2)] \times$$

$$\left\{ \frac{\sin[t T_K(v_1 + v_2)]}{v_1 + v_2} + \frac{1}{2} \cos[t T_K(v_1 + v_2)] \right\}. \quad (39)$$

Finally, we must handle the process $(c)$. After a bit of effort we find

$$t\frac{\partial}{\partial t} S^{(c)} = \frac{t T_K}{2\pi} \int_0^\infty \frac{dv_1 dv_2}{(v_1 v_2)^{1/4}} \frac{1 + v_1 v_2}{(1 + v_1^2)(1 + v_2^2)} \frac{v_1 - v_2}{(v_1^{1/2} + v_2^{1/2})^2} \exp[\varphi(\ln v_1)] \exp[\varphi(\ln v_2)] \times$$

$$\sin[t T_K(v_1 - v_2)]. \quad (40)$$

The only process which remains non-zero in the conformal limit is the sin term in (38): it is the "leading" contribution we have used to obtain the curve on Fig. 2. The other contributions are extremely tedious to evaluate numerically, because of the less favorable, highly oscillatory behavior of the integrals involved. (We also note that these integrals lead to naively diverging contributions in the IR, which have to be regularized using $\int_0^\infty dx e^{iax} \equiv \int_0^\infty dx e^{(ia-\epsilon)x} = -\frac{1}{ia}$). We have checked however that, while they do not add up to zero any longer, these contributions seem to remain relatively small throughout the crossover ($\lesssim 10\%$ for the points we were able to evaluate) but more work would be needed to investigate whether such contributions are cancelled by higher-order terms.

### A.3   The case of two weakly connected chains

We note that it is interesting to consider a more general quench between two systems with different, non-vanishing values of the coupling constant $\gamma$. The physics in this case is quite different, in particular, we expect that at large times the entanglement entropies differ by a finite constant. In the scaling limit, this function should depend only on the ratio of the two Kondo temperatures $T_K^{(1)}/T_K^{(2)}$:

$$S(T_K^{(1)}) - S(T_K^{(2)}) = F(T_K^{(1)}/T_K^{(2)}). \tag{41}$$

Mild analyticity assumptions then show that $F$ must be proportional to a logarithm. There are various ways to determine the proportionality constant. An amusing one is to use Form Factors again. To regulate things, we introduce a new scale $l$, that is we consider the entanglement of the wire going from $-\infty$ to $-l$ with the rest of the system. This entanglement is easily obtained using the one point function of the twist operators. Note that this time, the calculation is done in equilibrium, and no FF for boundary conditions changing operators are necessary. The steps are then described in Ref. [22]. One finds the leading contribution

$$S = -\frac{1}{8}\int_0^\infty \frac{d\omega}{\omega} e^{-4l\omega}\left(\frac{\omega}{T_K + \omega}\right)^2 + \dots \tag{42}$$

We can reformulate this into

$$S = -\frac{1}{8}\int_0^\infty \frac{d\nu}{\nu} e^{-4lT_K\nu}\left(\frac{\nu}{\nu + 1}\right)^2 + \dots \tag{43}$$

This is a finite quantity, function of $lT_K$. It goes to zero as $T_K$ (or $l$) goes to infinity as required physically. Note however that at $lT_K = 0$, it is logarithmically divergent.

It is possible to calculate similarly all the remaining terms. In the end, one finds that $S$ expands as an infinite sum of contributions with $2n$ particles, and that each of these terms has a singularity at high-energy (UV) of the form $g_n \ln lT_K$, while all the other terms are analytical. The term calculated in (43) $g_1 = \frac{1}{8}$. It is proven in [23] (see eq. (3.54), (5.17) and (5.10)) that

$$\sum_{n=1}^\infty g_n = \frac{1}{6}, \tag{44}$$

hence we find in the end, after letting $l \to 0$ so the contributions of all the analytical terms vanish, that

$$S(T_K^{(1)}) - S(T_K^{(2)}) = \frac{1}{6}\ln\frac{T_K^{(1)}}{T_K^{(2)}}. \tag{45}$$

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
