# Peer review of "Universal Entanglement Dynamics following a Local Quench"

_SciPost Physics, doi:SciPost Phys. 3, 001 (2017)_

## Round 1 · Referee Report · Olalla Castro-Alvaredo (Referee 1) · 2017-2-27

Strengths

1) The research carried out in this paper is very timely as it addresses a problem of much current interest, namely the time evolution of entanglement following a quantum quench.
2) The present work goes beyond the most commonly studied regime in this context, namely the evolution of entanglement at large times after a local quench. The latter was famously characterised by Calabrese and Cardy (in critical systems) and shown to scale logarithmically with time. In contrast, the present work considers the evolution of the entanglement entropy (a particular measure of entanglement) not only at large times but also for intermediate and small times. In this way it provides a full analysis of the evolution of the entanglement entropy (EE) from t=0 to t=infinity giving several interesting new insights into the evolution of EE a short time after the quench.
3) Related to the previous point, the current work provides both a numerical and an analytical study of the EE as a function of time. Although numerical studies of the EE after a quantum quench are relatively common nowadays there are very few analytical results going beyond the large time logarithmic scaling mentioned in 2). This is therefore a significant contribution even if the analytic results provided are not exact.
4) The analytical and numerical results are in very good agreement.
5) The analytical results although approximate follow from a non-trivial form factor computation which is presented in detail in the appendix.

Weaknesses

In my view the paper has one weakness. In order to compare the form factor (FF) calculation to the numerical results the authors use a “trick” which is really only justified by the fact there is good agreement with the numerical results. The authors have used this “trick” in a previous work, again finding good agreement with numerics. Because of this good agreement I think there is some justification in what they do but from the analytical point of view there is really no justification. Essentially by cutting the form factor expansion at the lowest non-trivial order, they find that the coefficient of the log t term that is predicted to describe entanglement at large t is not the expected 1/3 but rather ¼. As they point out, this is not surprising but rather a result of the truncation of the FF series which for t->infinity is slowly convergent. In other words further terms in the expansion would need to be added in order to approach the coefficient 1/3. I understand computing such terms would be very challenging and perhaps not feaseble at all. Instead the authors suggest that they may just multiply their FF result by 4/3 to get the correct large t behaviour. As I mentioned earlier, by doing this they indeed find a result that agrees very well with numerics as seen in Fig. 2. The problem with this is that there is really no reason why this should work so well. Besides, even if multiplying by 4/3 will certainly “correct” the large t behaviour it seems to me it should spoil the low t behaviour that I would expect is well described by the truncated FF expansion. Why not multiplying instead by say 4/3 t/(t+1)+sin(t)/t? This goes to 4/3 when t->infinity and goes to 1 when t->0. I know this would be a rather unusual guess but the point I am trying to make is that there are probably many ways of "renormalizing" the FF results which would have the right long and short time asymptotic and would fit the data well.

Report

I think the paper deserves to be published in SciPost. In my view its strengths are more prominent than its one weakness. Also I admit that dealing with this weakness is difficult and doing it rigorously would mean writing a rather longer and more technical paper. I think the good agreement between the analytical results and the numerical results provides good support for the work, even if some of the analytical results are not rigorously justified. The paper tackles a timely problem by various approaches and gives new insights into the time evolution of entanglement following a local quench. I think these new insights are very important and it would be nice to understand them in a more general context. For instance, a good question is how general are these results? Do they depend strongly on the model? Would the author’s trick of multiplying by 4/3 (or another convenient number, depending on the model) provide a good fit in all cases? I think these are all interesting questions and this paper paves the way to perhaps answering them in the near future. As such it is likely to have impact in the academic community working in this area.

Requested changes

Comments: I have a few comments regarding mainly small typos and one reference.

1) At the end of the first paragraph the authors refer to the special issue [2]. This special issue deals mainly with measures of entanglement rather than quenches. As far as I can remember there are very few papers (if any) that discuss quenches. On the other hand there is a more recent special issue published by JSTAT that focuses on systems out of equilibrium (see http://iopscience.iop.org/journal/1742-5468/page/extraspecial7). I think it would make sense to cite this instead of or in conjunction with [2]. 2) I have noticed that the word Rényi appears as “Reny” in at least two places. 3) I think it would make sense to refer to the appendix either just before equations (7)-(8) or just after. 4) In equation (9) the style of the superindices “a” is different in the second and third sum from the first sum. The same applies to the operators c_i, the coupling J’, U_1 etc. 5) After equation (9) I think it would be useful to remind the reader of what the operators c_i are. Similarly, it would be good to remind the reader of what the operator d in equation (2) is. 6) I was a bit confused by the statement in the Discussion section stating that: “Plotting the derivative emphasizes however the intriguing fact that the instant slope (wrt ln t) of the entanglement growth saturates at values greater than c/3 in the intermediate regime” It is obviously true that t S’ is not a monotonic function of t. However the EE itself is a monotonic function (since its derivative is always positive). Is it the case that the derivative of the EE is in general also monotonic? (hence this result is a surprise). Could you say a bit more about why this result is surprising? 7) Relating to my report, could the authors say anything more about their "renormalisation" by 4/3? Did they try other renormalizations? Do they expect other renormalizations to work even better? Do they know why this renormalisation works both for t large and small? Any additional discussion would be useful. 7) In the appendix I found three instances of the use of the word “disappearing” or “disappears” instead of “vanishing” or “vanishes”. For instance in the sentence after equation (12) it should say that “its moment and energy vanish”. In the same sentence “going” should be replaced by “go”. 8) Before or after equation (38) it would be useful to mention that the new variables v_i=u_i/Tk

---

## Round 1 · Referee Report · Anonymous (Referee 2) · 2017-3-15

Strengths

1) Manuscript deals with an interesting problem 2) Interesting analytical approach to calculation of entropy

Weaknesses

1) Some assumptions of the form-factor calculation should be properly clarified 2) Discussion/comparison of results to relevant previous work is missing

Report

The manuscript deals with entanglement evolution after a local quench
through an impurity, realized by the interacting resonant level model.
Namely, the system is composed of two free-fermion wires in their ground
states that are, at $t=0$, coupled to each other via an extra site.
In equilibrium, the value of the tunneling amplitude sets an energy
scale, the so-called Kondo temperature $T_K$, which governs the physics of
the problem. In the quench setting, the authors argue that the entropy
evolution should be governed by a universal crossover formula, depending
only on the variable $t T_K$. That is, for large times $t \gg T^{-1}_K$,
the chain appears to be healed and one should recover the CFT result
$S \sim 1/3 \ln (t)$. On the contrary, for small times the chains should
appear to be weakly coupled only, leading to a slower entanglement growth.

The problem is studied numerically on one hand, considering a lattice
model, and analytically on the other hand, using a form factor approach.
In particular, the entropy is calculated via the replica trick, rewriting
it as a time-dependent expectation value of a twist-operator, and
obtaining the leading order terms in a form-factor expansion.

In my opinion, the manuscript deals with an interesting problem,
and presents a new approach in handling it.
There are, however, some major issues that should be clarified
before it could be considered for publication. Moreover, the
presentation of the manuscript could also be improved.

Requested changes

1) The foremost concerning issue is the mismatch between the numerics vs. form-factor result. Namely, the main result (7) gives, in the $t T_K \gg 1$ limit, the prefactor 1/4, instead of the CFT result 1/3. The authors solve this problem simply by multiplying Eq. (7) with 4/3. This is a very questionable way of handling this mismatch, and the authors do not give any reasonable explanation. Why can one expect that higher order FF terms give just a multiplicative factor? How can one at all rely on the first order term, when the correction is so large? In fact, the agreement in Fig. 2 is rather qualitative. I believe, the authors should discuss this issue in much more detail.

2) It is somewhat unclear, why all the material about form-factor calculations is presented as an appendix. In the end, this is what is essentially new in this paper, numerical calculations of entropy were already presented in a previous work [7]. In my opinion, at least some of the RLM FF calculation should be moved to the corresponding section. In fact, right now this section is rather difficult to understand, as no details are given there.

3) The list of literature is rather one-sided. There are, in fact, a number of other works where the evolution of entanglement through an impurity was considered, see e.g.

EPL 99, 20001 (2012) J. Phys. A 46, 175001 (2013) PRB 91, 125406 (2015)

I believe, these results should be cited and discussed against the findings of the present manuscript.

4) There is a typo in the definition of Renyi entropy in the text before Eq. (4)

---

## Round 1 · Referee Report · Anonymous (Referee 3) · 2017-3-17

Strengths

1) Interesting analytical attempt at calculating the entanglement entropy in the crossover regime, which is a difficult problem. 2) Strong evidence for a universal scaling form of the entanglement entropy at intermediate times. 3) Variety of the techniques used: form factors, numerics, perturbed conformal field theory.

Weaknesses

I see only one weakness, but it is significant. As pointed out by the other two referees, the authors multiply their "first order" result by a factor 4/3 to correctly reproduce the conformal limit. I find it very hard to justify such a "trick", without knowing the higher order form factor contributions. Even more worrisome is the fact that the actual result of the first order expansion is not really shown in Fig.2. Near the end of the supplemental material, it is stated that several terms of the same order are discarded, because they agree less well with the numerics.

Report

This paper deals with the real time dynamics after connecting two wires through an impurity. In particular, the entanglement generated between the two wires is studied.

Overall the paper is interesting, and the subject is timely. The method used (form factor approach) allows to go beyond the known results of conformal field theory, and to explore the crossover physics at intermediate time scales of the order of the inverse Kondo temperature. The analytical results are also compared to numerical simulations.

My main unease with the manuscript as it stands lies in the comparison between their main result and the numerical simulations (see weaknesses). While reading through the text near figure 2, one gets the impression that the first order form factor calculation is shown, modulo multiplication by 4/3. This is not so if one believes what is written in the supplemental material: out of the five terms generated to first order, four are dismissed because they "give a result, which, in fact, agrees less well with the numerics". That such terms vanish in the conformal limit makes no difference; after all, what the authors are after is precisely the intermediate, non conformal, regime.

Requested changes

1) Show in Figure 2 the full first order result, not just equation (7). 2) State more clearly in the main text what is shown in figure (2).

It is crucial that those two points be successfully addressed. Even though the manuscript is interesting, it lacks clarity in the present form.

---

## Round 2 · Referee Report · Olalla Castro-Alvaredo · 2017-5-10

Strengths
The strengths of the paper are the same as I had already pointed out in my original report. The results are timely and interesting and there is very good agreement between the non-trivial analytical calculations and the numerics performed by the authors.
Weaknesses
In my original report I had found one main weakness relating the re-normalisation factor that was introduced seemingly ad hoc to achieve good agreement between a form factor calculation, the CFT prediction and the numerics. This weakness remains but the authors have made an effort to better explain where this prescription comes from and why it is expected to work.
Report
The authors have engaged constructively with all my original comments and they have made an effort to explain the origin of their form factor calculation normalisation, while being honest about the lack of a strong rational for this choice. I find both their answer and the changes they have introduced satisfactory. I consider that they have successfully addressed the points I had raised.
Requested changes
None
Author: Romain Vasseur on 2017-06-19 [id 146]
(in reply to Report 1 by Olalla Castro-Alvaredo on 2017-05-10)We thank the Referee for their useful comments and for recommending our work for publication in SciPost.
Author: Romain Vasseur on 2017-06-19 [id 145]
(in reply to Report 3 on 2017-05-24)We thank the Referee for their useful comments and for recommending our work for publication in SciPost. Concerning the "typo in the definition of Renyi entropy in the beginning of paragraph before Eq. (4)", we do not see a typo in this definition: would the referee prefer we define the Renyi entropy in a more standard way with a logarithm? We would be very grateful to the referee if he/she could clarify what specific aspect of this definition is a "typo".

---

## Round 2 · Referee Report · Anonymous · 2017-5-21

Strengths
The strengths of the paper are exactly the same as in my original report. The subject is timely, and interesting partial analytical results are provided on a difficult problem.
Weaknesses
1) Several not so small terms are dropped to make the agreement with the numerics better, and the authors fail to mention that important fact in the main text. This problem should be fixed.
2) The "renormalization" by a factor 4/3 is still a weakness, but it is discussed clearly and honestly now.
Report
In my previous report, which is quoted below, I requested the following changes:
* * *
1) Show in Figure 2 the full first order result, not just equation (7).
2) State more clearly in the main text what is shown in figure (2).
It is crucial that those two points be successfully addressed. Even though the manuscript is interesting, it lacks clarity in the present form.
* * *
Neither of the two changes has been made, I cannot recommend publication in the present form.
We are not talking about a minor detail here, but about the main statement of the paper. Equation (7), which is shown in figure 2, is the result of clear cherry-picking. The result to lowest order, as defined in the last paragraph on page 4, is a sum of five terms. Four (!) of them are simply discarded, as admitted to shortly after equation (38), (39), (40) at the end of the appendix on page 15. The procedure, designed to get better agreement with the numerical data, is not acknowledged in the main text. Hiding this crucial piece of information in the appendix is of course unacceptable, especially since the authors mention in the new version of the appendix that the discarded terms are not so small after all. [This is a separate issue from the "renormalization" by 4/3, the discussion of this prescription is now satisfactory.]
As a result, several statements made in the main text are misleading. For example, the authors explain the various physical processes contributing to lowest order on page 4, name the next paragraph "the result at lowest order", but show a result which is not the result at lowest order. At the beginning of the discussion several statements are also misleading, including again the use of "lowest order" which does not mention the discarded terms.
I also have a few comments on the part of the appendix mentioned above:
"The only process which remains non-zero in the conformal limit is the sin term in (38): it is the
contribution we have used to obtain the curve on Fig. 2. The other contributions are extremely
tedious to evaluate numerically, because of the less favorable behavior of the integrals involved
that are naively divergent without additional regularization. We have checked however that,
while they do not add up to zero any longer, they remain small ($\lesssim$ 10%) and can be essentially
neglected for our purpose."
1) The integrals are convergent, and do not require additional regularization. Please clarify.
2) If the authors can guarantee the contributions are of the order of 10%, it means they are able to evaluate them numerically, at least to some reasonable degree of approximation. Therefore, incorporating them in figure 2 as I requested in my previous report is possible. Otherwise please clarify, and mention these contributions in the main text.
3) 10% is not small, it is more than enough to significantly worsen the agreement shown in figure 2. Especially if these contributions happen to move the location of the maximum.
4) Is it 10% before, or after renormalization by 4/3?
Requested changes
The list of requested changes is exactly the same as in my previous report.
1) Show in figure 2 the full lowest order result, not just equation (7).
2) State clearly in the main text what is equation (7), and what is shown in figure 2. Failing to mention the discarded terms is not acceptable.
Author: Romain Vasseur on 2017-06-19 [id 147]
(in reply to Report 2 on 2017-05-21)
Referee 2 is concerned about the "leading contribution" plotted in the main text being "cherry-picked" among other terms. We do agree with the referee that the proliferation of "subleading" first order terms is one of the unsatisfactory aspects of this non-equilibrium massless Form Factor approach which would have to be clarified in future works. However, it is important to notice that the massless Form Factor expansion is not controlled by a small parameter anyway (in sharp contrast with massive theories for which the FF program is much more controlled). In that sense, it is equally (un)satisfactory to discard terms at first or second order in the expansion. Moreover, there is a clear sense in which the contribution we plot is the "leading" one, even though it is not the only first-order term: it is the only term that does not vanish in the IR limit (and is clearly "dominant" in that limit). We also checked that the contribution of the other terms is roughly one order of magnitude smaller. To be more accurate, we changed "lowest order" to "leading contribution" in the main text.
Incidentally, this contribution is the only integral that we were able to evaluate numerically in a satisfactory way (the other ones are less well-behaved in the IR, and/or are highly oscillatory), which is why we did not include the other contributions in Fig. 2. For the few points where we were able to compute these other contributions, we checked that they are relatively small as mentioned above, but are large enough to worsen the agreement with the numerical results. Although this is obviously unpleasant, the only way to clarify the situation will be to compute higher order contributions, and to find a stable way to evaluate these contributions numerically. Given that the first order calculations were already quite involved, we defer these investigations to future work.
Contrary to what Referee 2 seems to imply, we feel we have been especially honest about this point from the first version of our paper. We commented on these issues only in the appendix as we thought this would only be of interest to Form Factor experts, but following the referee’s recommendations, we were happy to add a few sentences in the main text as well. We also improved and clarified the discussion in the appendix.
Regarding the specific comments:
“1) The integrals are convergent, and do not require additional regularization. Please clarify.”
Some of these integrals are divergent in the IR. We clarified this sentence and added the explicit form of the IR integrals to illustrate this.
“2) If the authors can guarantee the contributions are of the order of 10%, it means they are able to evaluate them numerically, at least to some reasonable degree of approximation. Therefore, incorporating them in figure 2 as I requested in my previous report is possible. Otherwise please clarify, and mention these contributions in the main text.”
As we mentioned above, we were not able to evaluate these integrals in a satisfactory way, which is why we decided not to plot them. We were happy to clarify this point in the main text and appendix.
"3) 10% is not small, it is more than enough to significantly worsen the agreement shown in figure 2. Especially if these contributions happen to move the location of the maximum."
We agree, and as we clearly stated in the first version of our draft, these contributions do seem to worsen the agreement with the numerical results — even though we reiterate we could not find a way to evaluate these integrals accurately. We added a sentence to emphasize this point in the main text. However, we feel that one order of magnitude is enough to justify calling eq 7 the ``leading’’ contribution.
“4) Is it 10% before, or after renormalization by 4/3?”
10% refers to a relative error and is the same after and before renormalization.

---

## Round 2 · Referee Report · Anonymous · 2017-5-24

Strengths
Interesting problem and attempt of an analytical calculation of entropy
Weaknesses
4/3 deficiency in the form-factor calculation
Report
The authors have provided an explanation for the 4/3-renormalization, based on earlier work, where similar behaviour was observed. The argument is essentially that the form-factor calculation could only yield the ratio w.r.t the result in the conformal limit. Although there is no clear analytical justification behind this trick, it has also been used in earlier studies and miraculously yields rather accurate results. Clearly, it would be nice to obtain a better understanding of the mechanism behind this trick. However, the authors have now given an honest account of the state of affairs in the text, without overclaiming the utility of their method for the calculation of entropy. I believe that, despite the deficiency of the form-factor method, this is still a decent work with an interesting result and I recommend the publication of the manuscript.
Requested changes
Typo in the definition of Renyi entropy in the beginning of paragraph before Eq. (4) is still there and should be corrected

---

## Round 2 · Author Response

We apologize for the delay in getting back to you: both authors were traveling until recently.
We are now happy to submit a modified version of our manuscript which, we hope, will be acceptable for publication.
First, we thank the referees for their careful reading of the manuscript and their constructive criticism.
Since the three referees seemed essentially in agreement about the qualities and weaknesses of our paper, we will take the liberty to paraphrase their comments rather than cite each of them in turn.
In a nutshell, the referees found the paper interesting and timely, found that it contained interesting results, but complained about the ``brutal'' way we obtained our analytical curve by performing a seemingly arbitrary renormalization by a numerical factor ($4/3$) of the result of a form-factor calculation.
Before discussing this factor in detail, we would like to emphasize that we did not consider that the analytical calculation was the main point of the paper. Rather, we felt the most interesting result was the existence of a well defined scaling function describing the crossover of the entanglement entropy from $S=0$ to the conformal asymptotic behavior $S\sim {c\over 3}\ln t$ after a local quench involving a weakly coupled impurity. The existence of this scaling function is argued on general grounds in our paper, and amply verified by high quality computer simulations.
Coming to the analytical calculation presented in the paper, we also would like to emphasize that it is the {\bf only} calculation we are aware of that can produce any usable information about the scaling curve for ${dS\over d\ln L}$. Perturbative approaches have been shown, in this kind of problem, to either converge extremely slowly (and thus to be unable to produce useful results), or to be plagued with uncontrollable divergences. See for instance the paper arXiv:1305.1482 where some of these aspects are discussed in the related context of crossovers involving sizes instead of time. Meanwhile, particle propagation pictures ``\`a la Cardy Calabrese'' do not seem to be able to recover the kind of fine structure present in the crossover either. Hence, we believe our calculation, however unsatisfactory, has the merit of existing, and, by a strange coincidence which may well {\sl not be} a coincidence, does provide remarkable accurate results. This is why we decided to publish it, despite the shortcomings mentioned by the referees.
Now the main shortcoming is that, at the order of form-factors expansion we are working, we get a satisfactory result only if we multiply the result of our calculation, by a numerical factor $4/3$. While this may seem horribly ad hoc, there is in fact a rationale behind this. It originates from early calculations performed by F. Lesage and H. Saleur (J. Phys. A30 (1997) L457), themselves inspired by calculations of F. Smirnov. In a nutshell, what happens in many problems involving ``massless form-factors'' or form-factors in the UV limit, is that a) the integrals over rapidities are diverging and b) once these integrals are properly regulated, the expansion itself is divergent. The way to cure this second divergence is to focus on the {\sl ratio} of two quantities, for instance, in the case of the paper by Lesage and Saleur, the ratio of a correlation function evaluated at finite value of the impurity coupling and the same correlation function evaluated in the conformal limit. To put it schematically:
%
$$
R(T_B)\equiv {\hbox{FF expansion of} \langle ...\rangle (T_B)\over
\hbox{FF expansion of} \langle ...\rangle (\hbox{CFT limit})}=\hbox{well defined and convergent}
$$
%
The trick used in the paper of Lesage and Saleur is thus to calculate $R(T_B)$ using form-factors, and then multiply the result by the (known) result in the conformal limit to obtain the searched for result at finite $T_B$.
There are, to our knowledge, no strong results as to why this should work, especially because the form-factors involved in this kind of calculation are extremely complicated.
What we did in our paper is, in spirit at least, identical. Instead of multiplying by the numerical factor $4/3$, we could say that what we have done is perform a calculation of the ratio
%
$$
R(T_B)\equiv {\hbox{FF expansion of } S(T_B)\over
\hbox{FF expansion of } S(\hbox{CFT limit})}
$$
and then multiplied by the known CFT result. In this case, the form-factors expansion of the entanglement is itself in fact well defined (at the price of taking a derivative wrt t). The numerator goes as $S_{FF}\sim {1\over 4}\ln t$ at leading order, while $S_{CFT} \sim {1\over 3}\ln t$. Hence the overall ``renormalization'' by a factor $4/3$.
It would certainly be better to investigate in more detail the form-factors expansion, in order to see whether higher orders render this renormalization unnecessary indeed (by correcting the denominator into ${1\over 4}\ln t$). But this is an extremely technical endeavor, and, as we pointed out already, we felt it was not the main point of the paper.
We have, in the present modified version, explained the ``historical'' origin of the renormalization, and toned down our claim of doing an analytical calculation some more, so as not to confuse the reader into believing we accomplished more that we did. We have also, to answer the additional criticism of one referee, discussed a little more the behavior of the extra terms we had initially discarded. We have added comments emphasizing how more difficult the time dependent case is, compared with the equilibrium cases we had studied so far. Despite one of the referees' suggestion, we have preferred not to put the form-factors calculation in the main text, since a) it is not our main point and b) it is not that satisfactory anyhow. We have otherwise been happy to follow all the minor changes suggested by the referees.
We hope our manuscript will be accepted for publication.
Sincerely,
H. Saleur and R. Vasseur.

---

## Round 2 · List of Changes

1) We updated the abstract
2) We have explained the origin of the renormalization in detail in the main text and in appendix, and commented on the discarded terms
3) We made many minor changes and added references following the referees' recommendations

You are currently on this page

---

## Round 3 · Author Response

Dear Editor,

We are glad that referees 1 and 3 think that we have addressed their comments in a satisfactory way, and we thank them for recommending publication. Referee 2 is concerned about the ``leading contribution’’ plotted in the main text being ``cherry-picked’’ among other terms. We argue below (and clarified in the main text) why we think this contribution is indeed dominant, even though the other terms are hard to evaluate numerically. Following the referee’s recommendations, we also added comments in the main text and appendix about the discarded terms. We believe we have been very honest about the pitfalls and shortcomings of our massless form factor approach. We do agree with the referee that further work would be needed to understand higher-order term contributions, and to clarify why the massless Form Factor program appears to be less controlled in this non-equilibrium setup. However, we still believe it is remarkable that this simple ``leading’’ contribution can capture this non-perturbative crossover accurately.

We feel we have taken into account and answered, to a more than reasonable extent,  the referees’ criticisms, and hope the paper can be accepted  soon.

Sincerely,
H. Saleur and R. Vasseur.

---

## Round 3 · List of Changes

1) minor typos fixed throughout the paper
2) We added a discussion of the ``discarded terms’’ in the main text, both after eq 7 and in the discussion.
3) We clarified the appendix following Referee 2’s comments.

---

## Editorial Decision

published